# Wide Use of Hyaluronic Acid in the Process of Wound Healing—A Rapid Review

Magdalena Antoszewska, Ewa Maria Sokolewicz and Wioletta Barańska-Rybak *

Department of Dermatology, Venerology and Allergology, Medical University of Gdańsk, Debinki 7 Street, 80-952 Gdańsk, Poland; magdalena.antoszewska@gumed.edu.pl (M.A.); ewa.m.sokolewicz@gumed.edu.pl (E.M.S.)
* Correspondence: wioletta.baranska-rybak@gumed.edu.pl; Tel.: +48-58-584-40-10; Fax: +48-58-584-40-20

**Abstract:** Hyaluronic acid (HA), as one of the main components of the extracellular matrix (ECM), plays an important role in the process of wound-healing and tissue-repair processes due to its unique properties and different physiological functions. HA has an ability to maintain a moist environment that promotes healing, the stimulation of growth factors and cellular constituents, and the migration of various cells essential for healing. This paper offers a review of HA use in the process of wound healing, with emphasis on hard-to-heal wounds, and examines its various applications in ophthalmology and otorhinolaryngology. It proves HA to be a versatile agent which finds its use in various fields of medicine for its antioxidant, anti-inflammatory, antibacterial properties and accelerated wound healing.

**Keywords:** hyaluronic acid; hard-to-heal wounds; dermatology; ophthalmology; otorhinolaryngology; wound healing





## 1. Introduction

Hyaluronic acid (HA) is a natural polymer that consists of polyanionic disaccharide units of N-acetyl-D-glucosamine and β-glucuronic acid linked by β-1,4 and β-1,3 glycosidic linkages [1–5]. It is produced endogenously by the human body and is one of the main components of the extracellular matrix (ECM) [6]. HA is widely distributed, and it is a substantial constituent of many tissues, such as connective, neural, or epithelial, and the skin is the organ that contains most of the body's hyaluronan [1–3]. HA has a wide range of molecular weights, ranging from $2 \times 10^5$ to $10^7$ Da, and that can influence its physicochemical properties [7]. As opposed to other polymers from this group, HA is non-sulfated. It has multiple carboxyl and alcohol groups, along with a single amide group. The former is fully ionized at the extracellular pH, with the negative charge of HA subunits contributing to water attraction and retention [8]. HA is a hydrophilic molecule that is capable of holding up to 1000 times its weight in water. Notably, the hydrogen bonding occurs between adjacent carboxyl and N-acetyl groups [9,10].

Its osmotic activity is often described as non-ideal and disproportionately high to its molecular weight. This contributes to its profound effects on the distribution and movements of water and plays a major part in water homeostasis. Additionally, secondary hydrogen bonds form along the axis of HA, generating hydrophobic patches. Those in turn allow for association with other HYA chains, despite their negative charge, and extend its capability of nonspecific interaction with cell membranes and other lipid structures [11]. Various preparations of HYA for viscoprotection and viscosupplementation must be meticulously composed, as its viscous and elastic properties are directly correlated with its concentration and molecular weight. Both aforementioned properties depend on the rate of shear or oscillatory movement. Rapid movement reduces the viscosity and increases the elasticity, deeming it an ideal biological lubricant [12].

The biological activity of HA is due to its binding to different receptors. High-molecular-weight HA is responsible for the tissue hydration and stabilization of the extracellular matrix structure; furthermore, it also contributes to the osmotic balance. Moreover, it has anti-inflammatory properties. Low-molecular-weight HA has angiogenic activity but also can present pro-inflammatory activity [5]. Therefore, it can influence many physiological or pathophysiological processes, such as wound healing, inflammation, and angiogenesis.

HA plays a vital role in the process of wound healing. Because of various biological activities and different physiological functions, the interest in HA products has been constantly increasing. HA has an ability to maintain a moist environment that promotes healing and stimulates growth factors, fibroblasts, and keratinocyte proliferation. Due to its highly hydrophilic character, HA absorbs exudate and enhances cell migration [13]. It exhibits beneficial effects on wound healing, resulting in a decrease in inflammatory processes, and regulation of tissue remodeling, and HA shows an enhancement of angiogenesis [5,14].

There are several non-human methods to obtain HA—extraction from animal tissues, production from microorganisms, and cell-free methods using in vitro systems that utilize purified hyaluronidase enzymes (HAS) [15]. The former was discovered in 1930 and has been regarded as a traditional technique to acquire HA. Currently, however, the large-scale production of HA is associated with streptococcal fermentation. The product of Microbial HA production is also known as "vegan HA" or "HA from plant origin". Of the two microorganisms, Streptococcus equi and Streptococcus zooepidemicus, the former produces HA with a lower molecular weight than the latter [16]. Various economic evaluations indicate that the facility-dependent and labor costs have the biggest impact on the economic cost of this operation. Therefore, improving the fermentation titer and downstream yields and simplifying the downstream portion of the process could make HA production more affordable [17]. Cell-Free Methods of HA Production utilize Pasteurella multocida, which can produce HA in a cell-free environment. This has proven to be effective at the research scale; however, current cell-free systems cannot produce at large scales [18].

Many age-related health problems, such as arthritis, wrinkles, xerophthalmia, or intestinal disorders, are connected to reduced HA content in tissues [19]. The decline is caused by both decreased synthesis and increased degradation of the molecule in the body. Assessment of the effectiveness in maintaining adequate levels of HA has been a research target in many studies analyzing both exogenous HA supplements and probiotics promoting endogenous HA synthesis. HA is a polymer with a short half-life in vivo and is rapidly metabolized. Studies suggest that in order to preserve the long-term efficacy of HA in vivo, regular supplementation is necessary. Researchers believe that the promotion of HA synthesis carries more health benefits than the inhibition of HA degradation [19,20].

In recent years, there has been growing attention paid to HA products used in wound care, as they have shown promising effects. Therefore, the aim of our review is to summarize the latest reports on the use and effectiveness of products containing HA and its derivatives in the process of tissue repair and wound healing, mainly in the fields of dermatology, plastic surgery, ophthalmology, otorhinolaryngology, and aesthetic medicine, as well as dentistry and maxillofacial surgery.

## 2. HA in Selected Fields of Medicine

This rapid review on HA analyzes its impact on the process of wound healing, with emphasis on hard-to-heal wounds, and examines its various applications in ophthalmology and otorhinolaryngology.

### 2.1. Literature Selection

This rapid review addresses the use of HA in the process of wound healing. An initial review of the literature available on PubMed was conducted by two investigators (MA and EMS) independently, followed by a subsequent follow-up search to identify potentially

missed articles describing various forms of use of HA in the process of wound healing. Keywords and a search strategy were formulated, and the following search terms were used: ("hyaluronic acid" [MeSH Terms] OR "hyaluronic acid*" OR "HA" OR "hyaluronan" OR "hyaluronate" OR "hylan" OR "sodium hyaluronate") AND ("wound healing" OR "burn healing" OR "burns" [MeSH Terms] OR "burn*"OR "surgical wound*" OR "wound*" OR "chronic wound*" OR "nonprogressive wounds" OR "leg ulcer*" OR "skin ulcer*" OR "difficult to heal" OR "hard-to-heal" OR "difficult to manage wounds" OR "non-healing wounds" OR "diabetic ulcer" OR "diabetic foot ulcer" OR "DFU" OR "Diabetic Foot" [Mesh] OR "Varicose Ulcers" OR "Venous Stasis Ulcer*" OR "Venous Hypertension Ulcer*" OR "Venous Ulcer*" OR "Stasis Ulcer*" OR "tissue repair" OR "tissue regeneration"). Only English-language articles or articles with an official translation were selected. Both investigators extracted the data from the full texts of the selected literature and assessed the quality of the included literature independently.

### 2.2. HA in the Management of Hard-to-Heal Wounds

Hard-to-heal leg ulcers remain a major healthcare challenge and may severely compromise a patient's life [21,22]. The treatment of chronic wounds is not only difficult but also very expensive and consumes enormous amounts of dressings, bandages, and medical supplies [23]. Therefore, the main principle of treatment is to accelerate the healing process and shorten patients' hospital stays.

The wound-healing process includes four consecutive but overlapping phases: hemostasis, inflammation, proliferation, and remodeling [4]. In acute wounds, the healing process proceeds efficiently, but in nonhealing wounds, this process is adjourned in one of the phases due to certain physiological states, such as the infection or extensive size of wound, or underlying medical conditions [4,23,24]. Because of properties such as biocompatibility, biodegradability, bacteriostatic properties, and hydrophilic character, HA-based materials present a wide range of roles in the process of wound healing, such as restoring skin integrity, assuring a hydrated environment, encouraging skin re-epithelialization, and allowing the growth and migration of cells (keratinocyte, fibroblasts, etc.) to the wound bed; they are also used as drug delivery devices. HA has been used to produce different wound dressings, which are widely used in clinical practice. Many approaches of HA had been described in wound management, such as topical formulations, gauze, bandages, hydrogels, sponges, films, and other HA-based scaffolds, as well as skin substitutes (xenografts, allograft, and autografts) [13,25]. To enhance their therapeutic outcomes on wound healing, wound dressings can be enriched with many different bioactive agents [25]. There are no wound dressings that are suitable for all wound types; therefore, among various types of dressings, materials which would benefit the wound the most should be chosen.

One representative of traditional dressings is gauze. Among other kinds, including bandages and plasters, they can be considered passive wound dressings, which are used as primary or secondary wound dressings. They are responsible for protecting the wound from contamination, absorbing wound exudate, and stopping bleeding [25]. Dereure et al. reported that HA-impregnated gauze pads are safe to use for patients with ulcers of venous or mixed etiology. The improvement of the percentage of reduction of the wound, the percentage of complete healing, and pain intensity was shown, but the results were not statistically significant [26]. However, Humbert et al. reported that, in the local treatment of venous leg ulcerations, HA-impregnated gauze pads were significantly more effective than gauze pads without HA on wound size reaction, pain intensity, and number of healed ulcers. Treatment was well-tolerated by patients as well [27].

Novel approaches of HA-based dressings focus on modifying the physiology of the wound, provide a moist environment, and improve granulation and epithelialization, as well as on delivering various bioactive agents.

HA-based hydrogels are suitable to treat chronic wounds, such as leg ulcers, pressure ulcers, and diabetic foot ulcers (DFUs), as well as for the management of wounds that are prone to bleeding [28]. Hydrogels are 3D polymeric networks that are able to absorb

immense amounts of water and can provide a moist environment for cellular growth. Moreover, their porous structure allows cells, gasses, and nutrients to be accommodated easily [13]. Hydrogels are used to restore volume and hydration, as well as to correct scars, asymmetries, or defects of the soft tissue [29]. One of the commercially available hydrogels is Hylase Wound Gel®. It is a topical agent composed of a combination of emollients and sodium hyaluronate (2.5%). Its main strength is that it avoids tissue dehydration and supports the healing process.

For superficial wounds, films with HA can be used. Films are composed of transparent polymers that allow for oxygen and carbon dioxide exchange, permit water vapor transmission, and avoid bacteria penetration. One of the commercially available films is Hyalosafe®, which is a total benzyl ester of HA. It can be applied directly to the wound bed, and one of the greatest advantages of transparent materials is that wounds and the healing process can be constantly monitored without unnecessary dressing changes which may result in damage to the newly created epidermis. Films should be used for the management of wounds with moderate exudate, such as superficial surgical wounds and first- and second-degree burns [30,31].

Sponges like HylaSponge® absorb a large volume of water and assure the skin's hydration to promote wound-healing development. Sponges require secondary dressings or tapes/bandages to maintain at the wound site [32]. HylaSponge is used to treat acute and chronic wounds [31,33]. Sponge with HA in combination with zinc has been shown to reduce scarring in patients after bilateral breast surgery, which was associated with greater satisfaction of patients after treatment [34].

Laserskin® is a HYAFF biopolymer-based scaffold. This is a sheet of maximally esterified HA with perforations ranging from 40 μm to 500 μm, designed as a transfer mechanism for keratinocytes from tissue culture to wound bed. It is used in the management of burns and chronic leg ulcers [13,35].

Hyalomatrix® is a transparent membrane composed of HYAFF, and it is a flexible and bilayered dermal substitute. It is indicated for the management of partial- and full-thickness wounds, and it can be used for chronic wounds, second-degree burns, and surgical and trauma wounds, as it promotes wound closure, as well as dermis regeneration [13,30].

Taddeucci et al. evaluated a partial benzyl ester derivative of HA (Hyalofill-F®) [36]. It is a sterile, non-woven fleece wound dressing composed of HYAFF (Fidia Advanced Biopolymers, Turin, Italy). It is a rapidly hydratable material which interacts with fluids and exudates and transforms into a viscous clear gel. Thanks to the gel, a HA-rich tissue interface is created, which provides a moist environment that promotes the healing process and cell migration. It can be applied in the management of chronic wounds, including DFUs. When used in the treatment of venous leg ulcers in combination with compression bandaging, it was well-tolerated and may have a beneficial effect even in the treatment of hard-to-heal venous ulcers.

Graça et al. recently provided an overview of different types of dressings containing HA and its derivatives used in wound treatment [13]. In their review, some commercially available dressings were evaluated, and the shortcomings of HA alone were discussed. Chemical modification or cross-linking can overcome some disadvantages and shortcomings, improving its mechanical properties and biological properties [37]. In order to enhance HA stability and biological performance during the healing process, HA-grafted pullulan (HA-g-Pu) polymers were synthesized. Li et al. reported that HA-g-Pu polymers had better water stability and obtained a relative rapid hemostasis ability [38]. HA-based sponges, because of their weak mechanical properties, are being combined with other polymers, such as chitosan, alginate, or carboxymethylcellulose sodium, in order to improve the water stability and mechanical properties and biological features (hemostatic, antibacterial, and healing) [39,40]. Hydrogel based on cross-linked HA and chitosan showed not only good biocompatibility but also enhanced the cell migration and inhibited inflammatory cytokines production [41].

Persistent, concomitant infection in chronic wounds may significantly delay the healing process. Therefore, the antimicrobial and antiseptic properties of HA-dressings are being widely studied [42–44]. Zhou et al. obtained antibacterial effects of chitosan and HA-based hydrogel loaded with mupirocin in their in vivo study, showing promising results for the effective promotion of wound regeneration under complex infection conditions [42]. An in vivo study conducted by Chen et al. demonstrated that combined treatment of HA plus povidone-iodine may accelerate wound healing by enhancing wound epithelium, suppressing the inflammatory response, and facilitating cellular proliferation and angiogenesis [43]. Clinical improvement in patients using dressings with HA and Iodine Complex was observed in a group of patients with diabetic foot ulcers [44].

The development of HA-based scaffolds has focused on the enhancement of their biological performance, through the incorporation of bioactive molecules (like growth factors, natural product extracts, and sulfadiazine) and inorganic compounds [13,45]. De Angelis et al. reported stronger regenerative potential of scaffolds composed of HA and platelet-rich plasma in terms of epidermal proliferation and dermal renewal when compared with HA alone [46]. Kartika et al. evaluated the effect of topical use of autologous platelet-rich fibrin (A-PRF) and HA in DFU wound healing. The study was an open label, randomized, controlled trial conducted from July 2019 to April 2020 [47]. A combination of A-PRF + HA was applied on the surface of wounds and covered with bandage during the treatment period of 7 days. The results showed that adding HA to A-PRF has a positive effect on the formation of healthy granulation tissue and increases the concentration of the released growth factors. It results in a significant increase in VEGF, which results in increased angiogenesis. Moreover, a decrease in inflammatory progression was noticed. A combination of A-PRF and HA seems to provide beneficial effects in patients with DFU wounds.

Scaffolds loaded with HA and epidermal growth factor (EGF) result in thicker epidermis and dermis layers in vitro compared to scaffolds alone, so they can enhance wound healing [48]. Bioactive molecules have been incorporated into sponges' structure. Catanzano et al. incorporated tranexamic acid into composite alginate–HA sponge dressings to avoid excessive blood loss and promote hemostasis in patients after post-extractive alveolar wounds [49].

The topical application of HA can be used for the treatment of skin irritations and helps skin integrity restoration. HA delivered in a cream formulation significantly contributes to the restoration of physiologic conditions in wounds and promotes ulcer healing [50].

A 0.2% HA-based topical formulation (Ialuset cream; Laboratoires Genévrier, Paris, France) used in patients with chronic wounds of venous or mixed etiology resulted in a significantly greater reduction in wound size after 45 days of treatment with the HA cream compared with the control group. A reduction in pain intensity was reported [50].

Connettivina Plus® is a cream containing 10 mg/mL of hyaluronate sodium associated with argentic sulfadiazine that was effective for treating grade 2–3 pressure sores in patients with chronic ulcers [51].

HA in combination with polynucleotides has been recently studied [52–54]. The combination of HA and polynucleotides was demonstrated to be more effective than HA alone in promoting the re-epithelialization of venous ulcers [54].

Cassino and Ricci, in their prospective observational study, stated that the topical application of an amino acid dressing (composition of the powder and cream) may promote healing in venous, pressure, and DFUs. In total 160 patients with chronic wounds were analyzed, and 76% (n = 120) achieved a reduction in the ulcer size [55]. Abruzzese et al. tested gel formulation containing HA and amino acids in the treatment of neuropathic leg ulcers. They reported that the healing rate was significantly higher in comparison with groups treated with the inert gel vehicle and stated that the use of Vulnamin® gel is not only safe but also effective in the management of ulcerations due to the peripheral neuropathy in diabetic patients [53].

De Caridi et al. evaluated the topical application of Nucliaskin S™ (Mastelli s.r.l., San Remo, Italy), HA plus polynucleotides gel (PNHA), in patients with venous lower-limb ulcerations. After 45 days of the treatment, complete wound healing occurred in 60% of limbs, and the average area reduction was 67%. Their results strongly confirm that PNHA, via an elevated trophic effect, speeds the healing time of ulcers of the lower limbs [54].

Due to lack of local tissues and poor general conditions, the reconstruction of chronic ulcers is often impeded. There are several methods available for reconstructing dermis using HA. Porcine acellular dermal matrix and polynucleotides-added HA effectively promote wound healing and scar formation through epidermal stem cells and were previously reported to reduce hospitalization and promote cell proliferation and matrix production in skin ulcers [56–58]. In a single-blind randomized clinical trial, Segreto et al. proved that a combination of porcine dermis with polynucleotides-added HA is more effective than polyurethane foam in the treatment of chronic ulcers, not only clinically and histologically but also economically [56].

The use of a graft of an autologous fibroblast–HA complex for DFUs was proposed by You et al. [59]. The analyzed group consisted of 31 patients with ulcers on the borders between the dorsal and plantar sides. In 84% of patients from the studied group, complete healing was achieved. No adverse events were reported. Those results show promising options for patients with DFUs.

Another promising study was conducted by Çetinkalp et al., who evaluated medical device Dermalix (Dx), which is a collagen-, gelatin-, and laminin-based wound dressing containing resveratrol-loaded HA and dipalmitoylphosphatidylcholine-based microparticles [60]. Dx provided 2-times-faster wound healing and decreased oxidative stress. They also stated that Dx might be used as biodegradable and bioactive primary wound dressing for DFUs which are not infected. Proper treatment of chronic wounds due to diabetes mellitus may result in fewer hospital admissions and shorter hospital stays and may reduce morbidity rates and major limb amputations. The characteristics of Dermalix, as well as other dressing types, can be found in Table 1.

**Table 1.** List of some examples of HA-based wound dressings.

| Dressing Type | Commercial Name | Characteristics | References |
|---|---|---|---|
| HA-based hydrogels | Hylase Wound Gel® | ■ Absorbs immense amounts of water<br>■ Provides a moist environment for cellular growth<br>■ Porous structure allows cells, gasses, and nutrients to accommodate easily<br>■ Restores volume and hydration and correct scars, asymmetries, or defects of the soft tissue<br>■ Hylase Wound Gel® avoids tissue dehydration and supports the healing process | Khelfi (2018) [28]; Zerbinati et al. (2020) [29]. |
| HA-based films | Hyalosafe® | ■ Composed of transparent polymers<br>■ Allows for oxygen and carbon dioxide exchange, water vapor transmission<br>■ Avoids bacteria penetration<br>■ Wounds and healing process can be constantly monitored without unnecessary dressings changes | Longinotti (2014) [30]; Oro et al. (2011) [31]. |
| HA-based sponges | HylaSponge® | ■ Requires secondary dressings or tapes/bandages to maintain at the wound site<br>■ Absorbs a large volume of water<br>■ Assures skin hydration to promote wound-healing development<br>■ HylaSponge is used to treat acute and chronic wounds | Simões et al. (2018) [32]; Nguyen et al. (2019) [33]; Mahedia et al. (2016) [34]. |

**Table 1.** *Cont.*

| Dressing Type | Commercial Name | Characteristics | References |
|---|---|---|---|
| HYAFF hyaluronan-based biodegradable polymers | HYAFFbiopolymer-based scaffold — Laserskin® | ■ Sheet of maximally esterified HA with perforations ranging from 40 μm to 500 μm <br> ■ Used in management of burns and chronic leg ulcers | Price, Berry, and Navsaria (2007) [35]. |
| | HA-based membrane — Hyalomatrix® | ■ Transparent membrane composed of HYAFF <br> ■ indicated for management of partial- and full-thickness wounds <br> ■ Can be used for chronic wounds, second-degree burns, and surgical and trauma wounds <br> ■ Promotes wound closure and dermis regeneration | Longinotti (2014) [30]. |
| | Fleece wound dressing composed of HYAFF — Hyalofill-F® | ■ When applied at the wound site, it interacts with wound exudate, and a hydrophilic gel is produced <br> ■ Provides a moist wound environment that promotes cell activity and the healing process | Taddeucci et al. (2004) [36]. |
| Scaffolds composed of HA and platelet-rich plasma | / | ■ Scaffolds composed of HA and platelet-rich plasma in terms of epidermal proliferation and dermal renewal showed stronger regenerative potential when compared with HA alone | De Angelis et al. (2019) [46]. |
| Autologous platelet-rich fibrin (A-PRF) and HA | / | ■ Combination of autologous platelet-rich fibrin (A-PRF) + HA showed a positive effect on formation of healthy granulation tissue and increased the concentration of the released growth factors in patients with DFU <br> ■ Results in a significant increase in VEGF | Kartika et al. (2021) [47]. |
| Scaffolds with HA and epidermal growth factor (EGF) | / | ■ Result in thicker epidermis and dermis layers in vitro compared to scaffolds alone <br> ■ Can enhance wound healing | Su et al. (2014) [48]. |
| Tranexamic acid into composite alginate-HA sponge | / | ■ Avoid excessive blood loss and promotes the hemostasis in patients after post-extractive alveolar wounds | Catanzano et al. (2018) [49]. |
| Topical formulations (Cream, gels, and powders) | Ialuset cream® | ■ A 0.2% HA-based topical formulation <br> ■ Reduction in wound size and pain intensity | Dereure et al. (2012) [50]. |
| | Connettivina Plus® | ■ Cream containing 10 mg/mL of hyaluronate sodium associated with argentic sulfadiazine <br> ■ Effective for treating grade 2–3 pressure sores | Paghetti et al. (2009) [51]. |
| | Vulnamin® | ■ Powder, gel, or cream with HA and amino acids <br> ■ May promote healing in DFUs, venous and pressure ulcers | Abbruzzese et al. (2009) [53]; Cassino and Ricci (2010) [55]. |
| | Nucliaskin S™ | ■ HA plus polynucleotides gel (PNHA) speeds up the healing time of ulcers of lower limbs | Caridi et al. (2016) [54]. |
| Porcine dermis with polynucleotides-added HA | / | ■ Combination of porcine dermis with polynucleotides-added HA <br> ■ Promotes wound healing and scar formationPromotes cell proliferation and matrix production in ulcers <br> ■ More effective than polyurethane foam in the treatment of chronic ulcers | Segreto F et al. (2010) [56]. |

**Table 1.** *Cont.*

| Dressing Type | Commercial Name | Characteristics | References |
|---|---|---|---|
| Autologous fibroblast–HA complex | / | ■ In 84% of the study group, complete healing was achieved <br> ■ No adverse events were reported <br> ■ Promising option for patients with DFUs | You et al. (2014) [59]. |
| Dermal matrix | Dermalix (Dx) | ■ A 3-dimensional collagen–laminin porous-structured dermal matrix prepared and additionally impregnated with resveratrol-loaded HA and dipalmitoylphosphatidylcholine-based microparticles <br> ■ Provided 2-times-faster wound healing and decreased oxidative stress | Çetinkalp et al. (2021) [60]. |

### 2.3. HA in Ophthalmology

Progressively, more papers emerge on the importance of HA in various ophthalmologic procedures [61–65]. The authors of "Self-Cross-Linked Hyaluronic Acid Hydrogel in Endonasal Endoscopic Dacryocystorhinostomy (En-DCR): A Randomized, Controlled Trial", Yu et al., analyzed the healing process in a hundred and ninety-two patients affected with unilateral primary chronic dacryocystitis (PCD). The paper concluded that the agent may enhance the success rate of the procedure by promoting mucosal epithelial healing and preventing excessive granulation [66].

Corneal collagen cross-linking due to progressive keratoconus frequently leads to corneal abrasions. The implementation of HA-containing eyedrops improves healing by stimulating epithelial cell adhesion, proliferation, and migration [67–71]. A positive effect has also been achieved when adding trehalose to sodium hyaluronate, as trehalose offers notable benefits in the treatment of dry eye compared saline and hydroxy methylcellulose eye drops. As proven by Cagini, Carlo et al., 3% trehalose/0.15% sodium hyaluronate gel is effective in reducing DED symptoms after cataract surgery [72]. It was also proven to be superior to balanced salt solution after penetrating keratoplasty, as "only the Healon-treated group showed a high correlation to more complete graft healing one week postoperatively", stated by the authors, Reed et al. [73].

However, recent clinical studies state ReGeneraTing Agents (RGTAs) can improve corneal wound healing more effectively than HA, as described by Bata et al. in a randomized clinical trial [74]. Another prospective randomized study proposed to evaluate whether treatment with autologous serum exhibits better epitheliotropic properties than HA in diabetic patients proved the superiority of the former agent in regard to earlier closure of the corneal epithelium after intraoperative abrasion [75]. Controversies also arise as reports emerge correlating sodium hyaluronate with significant increase in intraocular pressure (IOP) in the early post-operative period after cataract surgery [76]. However, as Waseem et al. describe, an IOP rise can be observed with both 2% hydroxypropyl methylcellulose (HPMC) and 1% sodium hyaluronate (NaHa). Furthermore, a return to near-preoperative IOP values can be observed with both agents seven days after surgery.

Due to its ability to retain water, HA finds its use in ophthalmology as an additive immobilized directly into the lenses via chemical cross-linking or a releasable wetting agent or conditioning agent, the latter form used widely in contact lens solutions [77]. When added directly in the structure of contact lenses, it improves lens surface wettability, increasing biocompatibility of the lens with the eye and users' comfort [78].

### 2.4. HA in Otorhinolaryngology

The importance of HA in otorhinolaryngological and oral surgical procedures is steadily growing, despite its main association with dermatology and cosmetic surgery [79–85].

Already in the year 1987, a pilot study was published, deeming the treatment with HA an alternative to myringoplasty when treating small- and medium-sized dry perforations, excluding moist perforations. Furthermore, the paper describes the resulting scar as one

of 'normal' appearance [86–88]. Topical application of 1% sodium hyaluronate also aids the closure of tympanic membrane perforations, and in the majority of cases, the resulting scar resembles the normal tympanic membrane. According to a study by Kaur et al., out of 30 patients with dry central perforations, 86.67% responded positively to the treatment. The size of perforations was, on average, reduced by 86.49% [89].

Newer studies also underline the role of HA in otorhinolaryngology, especially as sodium hyaluronate (SH). The article "Pilot study on the effects of high molecular weight sodium hyaluronate in the treatment of chronic pharyngitis" investigates the efficacy and tolerability of exogenous high-molecular-weight SH. In their study, the authors, Leone et al., noted adequate compliance and the lack of adverse reactions, concluding SH to be effective and safe in patients with chronic pharyngitis [90].

An increasing number of studies describe promising results of the combination of SH and a pool of collagen precursor amino acids (AAs), namely glycine, L-leucine, L-lysine, and L-proline. An SH-AAs-based spray was included in a preliminary study analyzing the outcome of its use in patients with radio/chemotherapy-induced oral mucositis. The results of the preliminary data of an open clinic following twenty-seven consecutive patients with OM suggest accelerated healing of the lesion and reduction of pain. Furthermore, pain relief has been assessed as immediate, given it was noted 2 h from application. This suggests the agent's promising role as rapid and effective pain management, since aiding faster mucosal wound healing, especially when applied frequently [91,92].

However, its role as a pain-relieving agent is controversial. A study comparing secondary intention healing of oral soft tissues after laser surgery with and without the use of a compound containing amino acids and sodium hyaluronate concluded that, despite promoting faster healing by secondary intention in laser-induced wounds, the agent did not affect pain perception in the studied group [93]. Nonetheless, its role as a non-toxic, antimicrobial, anti-angiogenic and anti-inflammatory agent remains crucial, with the author Melrose underlining its role in promoting neural repair strategies [94–97].

In a double-blind randomized clinical trial, three different bioadhesive gels were evaluated in regard to microbial growth in the suture thread following post-surgical application of chlorhexidine gel, chlorhexidine–chitosan gel, and HA gel. The authors noted better healing rates in the group treated with chlorhexidine–chitosan gel, particularly in comparison with patients who used HA gel ($p = 0.01$). However, a microbiological analysis revealed that none of the agents allowed for a beneficial reduction in the populations of bacteria and fungi [98].

HA may also aid wound healing when enriched with vitamins and amino acids, as in vivo and in vitro studies suggest. Said combination yields promising results and is of great importance especially in patients affected with conditions that impair soft tissue healing. Preliminary results of a study conducted by Canciani et al. revealed favorable effects of the use of HAplus gel enriched with vitamins C and E, as increased vascularization and improved collagen fiber organization were observed [99]. Thanks to its regenerative properties, the intranasal use of sodium hyaluronate promotes faster recovery of the impaired ciliated cells in patients who underwent functional nasal surgery. Both mucociliary clearance and nasal mucosa regeneration have been said to be improved, according to Cassano et al. [100].

The role of HA emerges also as an elastoviscous agent aiding mucosal healing, thus allowing for better control of mucosal bleeding and limiting serious sequelae of endoscopic sinus surgery. In their paper entitled "Use of hylan B gel as a wound dressing after endoscopic sinus surgery", the authors describe promising effects of HA on stilling post-surgical bleeding and the prevention of postoperative adhesions and state a positive correlation between its application and mucosal healing as compared to the control group. However, the authors of the randomized control trial "The clinical effects of HA ester nasal dressing (Merogel) on intranasal wound healing after functional endoscopic sinus surgery" found no statistically significant difference when comparing HA dressings with a nonabsorbable packing requiring removal [101,102].

According to a study conducted by Gocmen et al., local injection of HA at 0.8% prolonged the bleeding time and increased hemorrhage and swelling in the early postoperative period after third molar extractions. Furthermore, there was no significant difference in regard to pain scored on the visual analog scale (VAS) and maximum interincisal opening between the group exposed to HA and the control group [103].

Interestingly, a randomized controlled split-mouth study was conducted, investigating the correlation between HA treatment and the outcome of post-extraction wound healing in patients with poorly controlled type 2 diabetes mellitus (DM2). In the course of the study, 0.8% HA was applied on sockets and wound closure rate (WCR), and clinical scores in wound healing scale (WHS) and pain intensity measured by the VAS score were studied. The site of extraction exposed to HA presented with an increased WCR of statistically significant difference. This yielded the authors, Marin et al., to conclude that HA, when placed in post-extraction sockets of patients with poorly controlled DM2, may be beneficial for wound healing [104].

Alveolar osteitis can be successfully managed with lyophilized HA combined with octenidine. The results of a clinical study analyzing the clinical outcomes in 58 patients highlight the efficacy with a statistically significant success rate of 96.0% after administration of a HA-based medical device [105]. As alveolar osteitis is considered to be among the common adverse sequelae of third molar extraction, the results of the study are of clinical importance/value.

Another study analyzing the efficacy of over-the-counter products on the healing of denture-induced ulcerations and patients' self-reported pain included seven treatment groups: denture grinding (control), topical application of corn oil gel (placebo), triester glycerol oxide gel, D-panthenol gel, D-panthenol mouthwash, L-arginine mouthwash, and HA gel. In their paper entitled "Effect of Over-the-Counter Topical Agents on Denture-Induced Traumatic Lesions: A Clinical Study", the authors, Bural et al., dismissed HA as a clinically significant alternative to the current protocol [106].

The various uses of HA also exceed space maintenance in non-grafted sinus lifting. Göçmen et al. compared the outcomes when using HA and Ultrasonic Resorbable Pin Fixation (URPF) in a split-mouth study. The clinical outcome was assessed based on the height of alveolar bone, reduction in sinus volume, bone density, and implant survival. URPF was proven to be superior to HA in regard to alveolar bone height and sinus volume reduction. Nonetheless, bone height was considered adequate to place implants on both sides in all patients [107].

HA finds its role also in peri-implant maintenance of immediate function implants. A pilot study was conducted, analyzing its efficacy in comparison with chlorhexidine. Despite both agents being deemed satisfactory for the preservation of the implant, de Araújo Nobre et al. suggest introducing a maintenance protocol, consisting of HA administration in the first two months and chlorhexidine between month two and six. This proposal is ascribed to/associated with the correlation coefficient between plaque and the bleeding index, suggesting a potentially better result for chlorhexidine at 6 months [108].

The main limitation of this rapid review is that our research group focused on the literature available on PubMed solely. Although follow-up search was conducted to minimize the number of missed articles, potentially valuable publications available on other databases were not taken into consideration.

Only English-language articles or articles with an official translation were selected, thus excluding a fraction of foreign-language articles on HA that appeared in the search.

## 3. Conclusions

In summary, HA is a versatile biomaterial with large applicability in various fields of medicine for its antioxidant, anti-inflammatory, antibacterial properties, and it accelerates healing [109–115]. Our review analyzed its wide range of applications, along with promising results of numerous studies, emphasizing its important role in the management of hard-to-heal wounds. Despite the encouraging results, more randomized studies are

needed to conduct systematic reviews and a proper meta-analysis, which in turn would allow for further conclusions, recommendations, and guidelines concerning use of HA in the process of wound management. Its recent application in other fields of medicine, such as gynecology, urology, and orthopedic surgery, are promising targets of research [116–119].

**Author Contributions:** Conceptualization, M.A. and E.M.S.; methodology, M.A. and E.M.S.; software, M.A. and E.M.S.; validation, W.B.-R., M.A. and E.M.S.; formal analysis, M.A. and E.M.S.; investigation, M.A. and E.M.S.; resources, M.A. and E.M.S.; data curation, M.A. and E.M.S.; writing—original draft preparation, M.A. and E.M.S.; writing—review and editing, W.B.-R., M.A. and E.M.S.; visualization, M.A.; supervision, W.B.-R.; project administration, W.B.-R., M.A. and E.M.S. All authors have read and agreed to the published version of the manuscript.

**Funding:** This research received no external funding.

**Institutional Review Board Statement:** The study did not require ethical approval.

**Informed Consent Statement:** Not applicable.

**Data Availability Statement:** No new data were created or analyzed in this study. Data sharing is not applicable to this article.

**Conflicts of Interest:** The authors declare no conflict of interest.

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
