# Peer review of "Wide Use of Hyaluronic Acid in the Process of Wound Healing—A Rapid Review"

_scipharm, doi:10.3390/scipharm92020023_

Round 1

Reviewer 1 Report

Comments and Suggestions for Authors

The manuscript "Wide Use of Hyaluronic Acid in the Process of Wound Healing―A Review" presents information on hyaluronic acid and its properties in ophthalmology and otorhinolaryngology. However, the authors could expand their research by adding:

- Add the structure of hyaluronic acid, highlighting the functional groups that are important for its action.

- Ophthalmology and contact lenses

- Otorhinolaryngology and tympanic membrane perforations

Author Response

We would like to wholeheartedly thank for your comments concerning our review. Changes have been made according to your suggestions, including but not limited to: 

Adding information on the structure of hyaluronic acid, its contribution to increasing the quality of contact lenses, and promising results, when used to treat tympanic membrane perforations. 

Best regards, 

Ewa Maria Sokolewicz 

Reviewer 2 Report

Comments and Suggestions for Authors

The use of hyaluronic acid in different fields of medicine has been known for a long time, especially in ophthalmology.

This work is based on a compilation of the use of this molecule and its derivatives in the healing of skin wounds, especially those of a chronic nature, and in the therapy of eye problems. The authors give us a current view of its use in wounds, where they provide a list of commercial products, trying to explain the possible mechanism of action, and other results that other researchers have obtained.

In my experience, although the potential of HA is great, its success rate in medical practice is variable. For example, in Table 1, you would include a column about its effectiveness (+++, ++, +).

Another issue that must be indicated in the introduction is what the non-human sources are or are to obtain and purify the HA, and the economic cost of this operation.

I don't know if the authors have found the number of water molecules that one HA molecule can absorb (lines 113, 130).

Author Response

We would like to wholeheartedly thank for your comments concerning our review. Changes have been made according to your suggestions. English language has been revised. Informations on obtaining HA have been added to the paper, as well as ways to make said operation economically more affordable. As we viewed the success rate of HA in medical practice to be very subjective, we decided not to include a column about its reported effectiveness in Table 1. We believe however, that more randomized studies are needed to conduct systematic reviews on said effectiveness in clinical practice, and believe it to be a promising target of research. 

Best regards, 

Ewa Maria Sokolewicz 

Reviewer 3 Report

Comments and Suggestions for Authors

Line 81 a citation without citation strange

Line 306 HA may also add. Sorry, but if the authors are not sure, why they prepared a review. The role of HA in wound healing is evident, but the authors need to read clinical trials and analyze them

Line 326 revise the grammar for “however” . It is contrast between two sentences, thus, it cannot be at the beginning of a phrase

Citations 83-85 They were analysed very superficially, the effect of Mw, and composition need to be included.  

Line 369, which kind of agent? Secret agent? The authors wrote “HA is a versatile agent”. It is necessary to use an adequate vocabulary for scientific publications. Definition of agent “a person who acts on behalf of another person or group”. I do not think this work is good to denominate a polysaccharide.

Line 372 “yield potential to new research possibilities” Wrong English

Yield produce or provide (a natural, agricultural, or industrial product) … strange for hyaluronan. Also, this is very general affirmation, also cellulose is import for fuels, lignin, starch. Then why to study HA. The authors do not know why and this article is not showing why.

Line 373 This is misleading the reading. The role of hyaluronan is lubrication, but it is not analgesic. This is very strange affirmation.

The conclusions are NOT CONCLUSIVE. It seems they were prepared by an undergraduate student from first year of medicine after reading several articles, without catching the idea.

Line 391.Citation 1 is quite old and irrelevant. The structure of HA is reported in many manuscripts.

Line 395. Citation 3 If the authors are revising wound healing why they are revising dry eye. Again irrelevant

Line 413. I would use a citation relevant in the field in the world scope and not only China, which is not so relevant and there are differences between the age and origin.

Several lines: It is a pity that the authors included only 3 citations about wound healing in diabetic ulcers, which is a very broad topic. One of these citations from the year 2009. I do not think that the authors prepared a good review. For example, they missed

Effectiveness of hyaluronic acid for treating diabetic foot: a systematic review and meta-analysis

10.1111/dth.12153

A Case Report of the Treatment of Diabetic Foot Ulcers Using a Sodium Hyaluronate and Iodine Complex

10.1177/1534734607304684

Line 588 a very general citation irrelevant as it is general knowledge.

Line 619 I do not think it is related to this topic.

I don’t believe that this review is really contributing in the art. There are many great reviews in the last decade about the topic, but this one is not contributing as the analysis is very superficial. Thus, no novelty and not contribution. I made some comments, but there are many wrong information in the paper that need to be re-analyzed by the authors.

10.3390/pharmaceutics14071479

10.1016/j.carbpol.2021.118006

https://doi.org/10.1016/j.carbpol.2020.116364

As a matter of advice, revise the last five years of literature to see what is new in the topic as HA is advancing really, fast. Revise the metabolism of the polysaccharide in vivo.

Comments on the Quality of English Language

There are many mistakes in the text, which I am not going to point out but this review needs proof-reading to use adequate verbs, adverbs and grammar.

Author Response

We would like to wholeheartedly thank for your comments concerning our review. Changes have been made according to your suggestions. English language has been revised. We added information on the structure of hyaluronic acid and its mode of action, its contribution to increasing the quality of contact lenses, and promising results, when used to treat tympanic membrane perforations. 

Furthermore, informations on obtaining HA have been added to the paper, as well as ways to make said operation economically more affordable. 

We realize that DFU patients are a huge group of patients for whom the use of HA-containing products may also be beneficial, but because of our goal, which was mainly to review the different forms of HA available for wound care, we did not focus separately on this group, which does not mean that we consider it less important. Nevertheless, we did our best to add more information on HA in products for DFU patients.

We hope that this review of different HA preparations and their use in clinical practice is now more readable and comprehensive. 

Best regards, 

Ewa Maria Sokolewicz 

Reviewer 4 Report

Comments and Suggestions for Authors

The submitted manuscript offers a thorough review of HA-based wound dressing materials/devices in medicine and surgery, effectively summarizing product categorization and clinical study findings. However, a notable limitation is the lack of comprehensive information regarding the mechanism of action of each product/material. Understanding the mode of action is pivotal for optimal clinical utilization. Merely listing products and study results may not suffice. Therefore, I recommend that the authors consider a complete restructuring and rewrite of the manuscript, including a detailed summary of the biological role of HA-based wound dressing materials. Additionally, the inclusion of schematic illustrations could enhance clarity and understanding for readers.

Author Response

We would like to wholeheartedly thank for your comments concerning our review. We added information on the structure of hyaluronic acid and its mode of action, its contribution to increasing the quality of contact lenses, and promising results, when used to treat tympanic membrane perforations. 

Furthermore, informations on obtaining HA have been added to the paper, as well as ways to make said operation economically more affordable. We hope that this review of different HA preparations and their use in clinical practice is now more readable and comprehensive. 

Best regards, 

Ewa Maria Sokolewicz